# The Importance of Offering Exome or Genome Sequencing in Adult Neuromuscular Clinics

**DOI:** 10.3390/biology13020093

**Published:** 2024-02-02

**Authors:** Laynie Dratch, Tanya M. Bardakjian, Kelsey Johnson, Nareen Babaian, Pedro Gonzalez-Alegre, Lauren Elman, Colin Quinn, Michael H. Guo, Steven S. Scherer, Defne A. Amado

**Affiliations:** 1Department of Neurology, University of Pennsylvania, Philadelphia, PA 19104, USA; 2Sarepta Therapeutics Inc., Cambridge, MA 02142, USA; 3Spark Therapeutics, Inc., Philadelphia, PA 19104, USA

**Keywords:** neurogenetics, genetic testing, genetic counseling, neuromuscular

## Abstract

**Simple Summary:**

There are many potential benefits of obtaining a genetic diagnosis for an adult diagnosed with a neuromuscular disorder. However, not all patients with neuromuscular disorders will be offered broad genetic testing to identify a genetic cause. Here, we use five case examples to illustrate why it is important for adult neuromuscular clinics to offer broad-based genetic testing to their patients. For each case example, we describe why broad genetic testing was better suited to detect the genetic cause of the condition, including a description of the genetic change identified and the resulting biological implication, as well as the clinical and personal outcomes of the genetic diagnosis for the patient. Our findings highlight the crucial role of broad genetic testing in the management of adults with neuromuscular disorders.

**Abstract:**

Advances in gene-specific therapeutics for patients with neuromuscular disorders (NMDs) have brought increased attention to the importance of genetic diagnosis. Genetic testing practices vary among adult neuromuscular clinics, with multi-gene panel testing currently being the most common approach; follow-up testing using broad-based methods, such as exome or genome sequencing, is less consistently offered. Here, we use five case examples to illustrate the unique ability of broad-based testing to improve diagnostic yield, resulting in identification of *SORD-*neuropathy, *HADHB*-related disease, *ATXN2*-ALS, *MECP2* related progressive gait decline and spasticity, and *DNMT1*-related cerebellar ataxia, deafness, narcolepsy, and hereditary sensory neuropathy type 1E. We describe in each case the technological advantages that enabled identification of the causal gene, and the resultant clinical and personal implications for the patient, demonstrating the importance of offering exome or genome sequencing to adults with NMDs.

## 1. Introduction

Neuromuscular disorders (NMDs) are a heterogenous group of conditions that impact the peripheral nervous system or muscle and can be characterized by features including weakness, sensory loss, contractures, and/or extraneurologic manifestations [1,2]. Genetic testing has become an integral component of care in patients of all ages with NMDs, especially with the increasing number of clinical trials and approved therapies that are gene-specific [1,3].

Historically, single gene testing was utilized due to cost and test availability, and may still be used when a particular diagnosis is suspected based on the phenotype or prior workup [3]. Subsequently, multi-gene panels curated by phenotype have become the most common methodology for genetic evaluation of NMDs, given their wide availability and affordability [3]. While panel testing has greatly improved diagnostic yield, there are significant limitations to this approach, many of which can be addressed by more advanced genetic testing technologies. It is therefore important that providers understand the potential benefits and limitations of these techniques. Two broad genomic sequencing tests, exome sequencing (ES) and genome sequencing (GS), have unique advantages over many other testing types such as single gene or panel testing, but are not yet integrated into routine clinical practice in many institutions [3]. ES uses targeted capture probes that hybridize and pull down the coding sequences of genes across the genome, thus enriching the sequencing for protein-coding regions. In contrast, GS does not perform targeted capture and provides more uniform sequencing coverage, provides information on noncoding regions, and is better able to detect small structural variants [4]. Although these tests are often referred to as “whole” exome and genome sequencing (WES and WGS respectively), this is increasingly recognized as a misnomer, as neither the sequencing nor the analysis covers the entire dataset [5,6].

Arguably, the most important benefit of genetic diagnosis is the potential to treat or enroll the patient in a clinical trial. Additional potential clinical benefits include diagnostic/prognostic clarification; more appropriate surveillance, particularly for extraneurologic complications; and the avoidance of additional workup that may be invasive, costly, or time consuming. For example, a retrospective analysis of a cohort of patients with NMDs who underwent exome sequencing identified extensive prior years-long workup, including muscle or nerve biopsy, electrodiagnostic testing, or other nondiagnostic genetic testing [1]. The benefits of diagnosis go beyond clinical utility and include the psychological impact of providing an answer for what caused a person’s condition, access to more individualized support groups and other resources, the empowerment of patients to make informed reproductive planning choices, and the provision of risk information for family members.

The diagnostic yield of genetic testing in patients with NMDs varies based on genetic testing strategy as well as the phenotypic spectrum of the cohort, with the highest diagnostic rates typically being reported in patients with muscle and myasthenic conditions [7,8]. This indicates that there is room for improvement in both the practice of test offering and the choice of testing strategy, and suggests the potential for additional new gene-disease association discoveries [8]. Given the numerous potential benefits of genetic diagnosis, we feel it is imperative that comprehensive genetic testing options be considered as part of routine clinical care for adult patients with NMDs of unknown cause.

For each illustrative case below, seen in our Penn Neurology and Neurogenetics clinics, we will discuss: (1) why ES or GS enabled the discovery of a genetic etiology and what the diagnosis may teach us about the gene identified; and (2) how the genetic diagnosis achieved through ES or GS impacted the patient, including therapeutic intervention/clinical trial access, reproductive/hereditary implications, ending the diagnostic odyssey, and/or providing closure and psychosocial support for patients and their families. For each case example, all genetic testing was completed in a commercial laboratory that is Clinical Laboratory Improvement Amendments (CLIA) certified as is the standard for clinical genetics care in the United States. Though laboratories use different pipelines for sequencing analysis and variant filtering, all laboratories use standardized guidelines for variant interpretation [9].

## 2. Detailed Case Descriptions

### 2.1. Case 1

The proband initially presented at age 20 for neurologic evaluation after experiencing progressively slowed gait and bilateral foot drop starting at age 10. At the time of evaluation, she was no longer able to stand for more than fifteen minutes, could not open jars or write, and could not feel temperature up to her waist. Her seventeen-year-old brother was evaluated in parallel; he was a cross-country runner who had gone from being the fastest to the slowest on his team, and also had sensory loss and hand weakness. EMG testing of both siblings revealed severe motor much greater than sensory axonal polyneuropathy. Both additionally had attention-deficit disorder, depression, and joint laxity, and their father also had joint laxity. Her brother was the first to undergo genetic testing via a hereditary neuropathy panel, which revealed a heterozygous variant of uncertain significance (VUS) in *TRPV4* (c.1392C>T, p.Arg464Arg), but the proband was not found to have this variant. She then underwent proband-only exome plus mitochondrial sequencing that was negative.

Three years later, she underwent exome reanalysis, including samples from her parents and her brother, revealing biallelic variants in *SORD*, one pathogenic and the second likely pathogenic (c.757delG p.Ala253QfsX27, frameshift variant in exon 7 (NM_003104.5), inherited from mother; c.458C>A, p.Ala153Asp, missense variant in exon 5 (NM_003104.5) inherited from father). These biallelic variants were also present in her brother.

The *SORD* gene [OMIM: 182500] encodes the second enzyme of the polyol pathway, sorbitol dehydrogenase, that oxidizes sorbitol into fructose, and function-disrupting variants lead to increased intracellular sorbitol. This in turn leads to selective degeneration of peripheral axons through yet-unknown mechanisms [10], causing autosomal recessive hereditary neuropathy that is often motor-dominant. In our proband and her brother, sorbitol levels were found to be elevated. They were subsequently enrolled in a therapeutic clinical trial.

#### Discussion

Genetic Diagnosis & Technology Considerations

There are several reasons that ES reanalysis was able to identify a genetic diagnosis in this case, when panel testing could not. First, variants in *SORD* are difficult to detect due to confounding by the homologous *SORD2P* pseudogene [10,11]. Pseudogenes are noncoding regions of the genome that resemble functional genes but have coding-sequencing deficiencies; when a pseudogene is present it can complicate variant interpretation as it may interfere with variant detection in the genuine gene [12,13]. In fact, although pathogenic variants in *SORD* are now believed to be the most frequent cause of recessive inherited neuropathies, they were not identified until 2020 since sequencing analysis pipelines missed *SORD* variants [10,11]. In this case, ES technology was able to overcome pseudogene confounding as a result of higher sequencing coverage, allowing for more reads uniquely aligned to the gene rather than the pseudogene [13,14]. GS often provides even greater discriminatory ability between genes and pseudogenes, due to inclusion of intronic sequences, which are more likely to be divergent, and would have also been a good testing option here. Since the discovery of *SORD* neuropathy, most patients have been found to harbor the c.757delG variant in either a homozygous state or in a compound heterozygous state with a likely pathogenic variant [10,11,15], including the above cases. The other variant identified in this family, c.458C>A, has also been observed in the published literature, though sometimes without segregation studies available to determine whether the variants were in trans [10,11,15].

Second, the ability to incorporate parental samples in ES greatly improves diagnostic yield by allowing for segregation analysis, which improves the quality of variant interpretation by clarifying the phase of the variants (as being in *cis* or *trans*, meaning on the same or opposite alleles respectively) and/or prioritizing de novo variants for individuals without family history [16].

Third, once sequencing is complete, most commercial laboratories store the sequencing data and offer an opportunity for periodic reanalysis as new discoveries are made. The ability to complete reanalysis was critical to this case, as the discovery papers for *SORD* neuropathy were published between the initial sequencing and reanalysis.

2.Therapeutic Intervention & Personal Impact

The siblings were enrolled in a clinical trial (NCT05397665) testing the pharmacodynamic efficacy and clinical benefit of AT-007 (gavorostat), an aldose reductase inhibitor that has been shown in preliminary studies to significantly reduce plasma sorbitol levels. In this case, a genetic diagnosis enabled treatment of what would otherwise have been called an idiopathic Charcot-Marie-Tooth (CMT), for which symptom management would be the only approach available. Therapies targeted to *SORD* neuropathy will likely evolve and benefit from future studies in preclinical models such as the recently developed *Sord*^−/−^ rat, which develops a motor-predominant neuropathy that mimics the human phenotype [17].

The parents of the proband were determined in their quest to identify the cause of their children’s neuropathy, and this diagnosis ended the diagnostic odyssey. This result also clarifies the chance of recurrence if the proband or her brother were to have children in the future, and it allowed other family members to be informed of their potential carrier status.

### 2.2. Case 2

The proband first came to neurologic attention at one year of age after failing to meet motor milestones. An EMG at that time revealed an axonal polyneuropathy. Toe-walking was noted at age four and hand weakness developed at age eight. At age 13 he was re-evaluated after having developed bilateral ptosis, intermittent exotropia, spasticity of conjugate gaze, a transverse smile, and tongue fasciculations, with now minimal use of his hands and feet, though he remained ambulatory with the use of orthotics. He was told at that time that he likely had a form of CMT. Around this time, his older brother, who had a similar but less severe condition, passed away at age 20 from respiratory failure that developed acutely after lifting weights. At age 23, the proband had sudden onset of severe soreness and hematuria along with profound weakness including respiratory failure requiring mechanical ventilation. CK reached 10,000 with elevated transaminases. He ultimately recovered and was discharged from the hospital on BiPap. A subsequent quadriceps muscle biopsy showed chronic neurogenic atrophy. Genetic testing was sent in 2004 including a 7-gene CMT panel, *PMP22* duplication/deletion testing, and a Friedreich’s ataxia repeat expansion test, all of which were unrevealing. He had a total of five more episodes of dark urine and severe soreness, often with weakness that occasionally involved respiratory function.

In 2017, the proband sought neurologic evaluation at age 48, after more than 10 years without neurologic care and 13 years since last genetic testing, to inquire whether novel genetic discoveries could inform his course. ES was obtained and revealed compound heterozygous likely pathogenic variants in the *HADHB* gene (p.W420G, c.1258 T>G in exon 15 (NM_000183.2) inherited from father; p.R229Q, c.686 G>A in exon 9 (NM_000183.2), inherited from mother). At the time of testing, neither variant had previously been reported in the literature, nor were they present in significant frequency in large population cohorts.

The *HADHB* gene [OMIM: 143450] encodes the beta subunit of the mitochondrial trifunctional protein (MTP). This subunit is part of a catalyst for three steps in the beta-oxidation of fatty acids, with the beta subunit containing 3-keto acyl-CoAa thiolase activity [18]. Loss of *HADHB* function leads to MTP deficiency, a rare autosomal recessive disorder affecting long-chain fatty acid oxidation. Affected patients can develop any of three syndromes: early onset cardiomyopathy and early death; recurrent hypoketotic hypoglycemia; or, as in the case of our proband, a sensorimotor neuropathy with episodic rhabdomyolysis [19], all exacerbated by fasting or intake of long-chain fatty acids.

The proband was instructed to avoid high-fat foods and fasting, and to follow a specialized diet including medium-chain triglycerides and MCT oil. He subsequently lost significant weight and noted that he was able to ambulate further and with marked reduction in fall frequency, and he also stopped fatiguing when chewing. He established care with cardiology and ophthalmology, and no heart or eye abnormalities were found. On pulmonary evaluation he was found to no longer need his BiPap, though he continued using it for comfort. He was admitted to the hospital in early 2020 with respiratory failure that was felt to be related to his underlying condition and subsequently passed away at age 50.

#### Discussion

Genetic Diagnosis & Technology Considerations

At the time of the proband’s last unsuccessful round of genetic testing in 2004, the association between recessive *HADHB* mutations and neuropathy with recurrent rhabdomyolysis was a novel finding that had only recently been reported [19], although the association of this phenotype with recessive *HADHA* mutations was reported as early as 1998 [20]. *HADHB* was not a part of the panel testing in 2004, presumably since gene panels are curated to include only those genes commonly implicated for a specific phenotype, such as CMT or myopathy, and there is a lag between the discovery of new genetic causes and their incorporation into panels. *HADHB* is now a part of several panels, including those for CMT, metabolic myopathy, rhabdomyolysis, and fatty acid oxidation. This case demonstrates the ability of ES to capture rare or newly discovered genetic conditions that may not be included in panel analyses.

2.Therapeutic Intervention & Personal Impact

In this case, the genetic diagnosis enabled the proband to make dietary and behavior modifications that had a positive impact on his health and quality of life. It also motivated a search for extraneurologic manifestations that could have endangered his heart and vision. If ES had been performed earlier, the dietary and behavioral modifications could have been even more impactful, and some costly and invasive procedures for the patient (including a muscle biopsy, many EMGs, and inconclusive genetic testing) could have been avoided.

The genetic diagnosis provided answers to the patient and his family more than 40 years after symptom onset, as well as answers regarding his deceased brother. After the patient’s passing, his mother described a sense of peace since her family had had an opportunity to understand their sons’ condition, which had impacted them greatly, and that her son had lived relatively long and well for someone with his disorder. In his case, she was also grateful for the management changes that accompanied his diagnosis.

### 2.3. Case 3

The proband was evaluated at age 50 with approximately one year of twitching in her left arm and leg that was initially diagnosed as benign fasciculations. On initial exam, she had diffuse hyperreflexia and mild left hand weakness. She subsequently developed left arm weakness, and after an EMG, she was diagnosed with amyotrophic lateral sclerosis (ALS). There was no known family history of ALS or related conditions. Next-generation sequencing panel testing and *C9orf72* repeat expansion analysis were performed by an external care team and were negative. In discussing sending additional genetic testing, we learned that the proband’s 83-year-old mother had developed progressive changes in her speech over the last 5 months. Her neurologic evaluation revealed lower motor neuron signs in bulbar and lumbosacral regions as well as upper motor neuron signs in these and cervical regions, and she, too, was diagnosed with ALS. GS was sent for the mother-daughter duo. The mother was found to be homozygous for an intermediate expansion of 29 repeats in *ATXN2*, and the daughter was found to be heterozygous for the intermediate expansion of 29 repeats in *ATXN2*. Both the proband and her mother have since passed away from ALS.

There are no other family members known to be affected with ALS, and there is no reported consanguinity in the family. Therefore, it is unusual that a homozygous intermediate expansion was identified in the proband’s mother; homozygous expansions are more rarely reported in the literature [21]. The proband’s sibling was counseled that they are presumed to be at least heterozygous for an intermediate expansion. Currently, the sibling is asymptomatic and has chosen not to undergo genetic testing, as this is a risk factor with limited data on penetrance.

The *ATXN2* gene [OMIM: 601517] encodes for ataxin-2, an RNA binding protein that modulates translation of many transcripts and regulates stress granule assembly [22,23,24]. Polyglutamine (CAG) expansions of >34 repeats in *ATXN2* have been known to cause spinocerebellar ataxia type 2, but more recently intermediate expansions have been found to increase risk for ALS [25]. Ataxin-2 was identified as a modifier of TDP-43 toxicity, directly linking it to the pathophysiology of ALS, with initial estimates showing a frequency of intermediate expansions in 4.7% of cases of ALS [22,25]. In at least one individual, an intermediate expansion caused symptoms of both SCA2 and ALS [26]. However, there is disagreement in the literature regarding the threshold at which repeats impart risk of ALS, with the initial publication finding 27–33 repeats to be higher in patients with ALS than controls [25], and later reports claiming that 29 [21,27,28], 30 [29], or 31 [22] or greater repeats are needed to confer increased risk.

#### Discussion

Genetic Diagnosis & Technology Considerations

Microsatellite repeat expansions are a common variant type in genetic neurodegenerative conditions. For example, trinucleotide repeats in *HTT* cause Huntington’s disease, hexanucleotide repeat expansions in *C9orf72* are the most common genetic cause of ALS and frontotemporal dementia (FTD), and other repeat expansions cause some forms of familial spinocerebellar ataxias [30]. While panel testing is often used as a first-tier test due to fast turnaround time or lower cost, repeat expansion analysis is often not included in panel testing, or in ES. Although GS can detect repeat expansions, such as in our proband and her mother, not all laboratories are validated for repeat expansion analysis of their GS data. Laboratories that do include repeat expansion analysis may only do so for a validated subset of genes. It is therefore critical to review the methodology of a laboratory’s GS prior to ordering testing if repeat expansion disorders are on the differential.

Intermediate expansions in the *ATXN2* gene were first described as a risk factor for ALS in 2010 [25], and *ATXN2* testing was not routinely included in clinical genetic testing for ALS until 2021. The initiation of a clinical trial of intrathecal injection of BIIB105, an antisense oligonucleotide targeting ataxin-2 (NCT04494256), has motivated more routine incorporation of *ATXN2* testing in the ALS genetic workup. Since few commercial laboratories include *ATXN2* repeat expansion analysis as part of their ALS panels, it is usually sent as an additional, single-gene test. In this case, GS was ordered from a commercial laboratory validated for *ATNX2* repeat expansion analysis, which detected the intermediate expansion where panels or ES could not. GS also permitted comparison of the findings of the mother-daughter duo in a single test.

2.Therapeutic Intervention & Personal Impact

Participation in the clinical trial for *ATXN2-*ALS was raised as a consideration for the proband and her mother, but neither patient met inclusion criteria owing to their advanced disease at the time of diagnosis. In a rapidly progressive disorder like ALS, an early genetic diagnosis is critical to maximize chances that a patient will meet inclusion criteria for trials. Additionally, for therapies that slow progression of a disease, earlier initiation of therapy should lead to better outcomes, as in the case of *SOD1*-ALS [31]. GS provides a means of broad and maximally inclusive coverage with a single test.

Although in this example the genetic diagnosis did not result in changes in medical management, it did provide an answer as to why both the proband and her mother developed ALS. This information was communicated to family members, who were given the option of genetic counseling and predictive genetic testing.

### 2.4. Case 4

The proband first presented for neurologic care at age 49, with about 5 years of progressive gait disturbance. Although she did not have any formal diagnoses, the patient shared that her neurodevelopmental history included learning differences and distractibility in elementary school, and a history of breath-holding spells with loss of consciousness when feeling fearful as a child. On exam, she was hyperreflexic with spastic tone of all extremities. Her family history was unremarkable. She had previously completed panels for hereditary spastic paraplegia (HSP) and neuropathy with nondiagnostic results. When she presented to our clinic, her evaluation was most suggestive of HSP or primary lateral sclerosis. ES identified a heterozygous variant: *MECP2* c.1164_1188del p.P389Rfs*12 in exon 4 (NM_004992.3), interpreted as a pathogenic variant.

The *MECP2* gene [OMIM: 300005] encodes for methyl-CpG binding protein 2 (MeCP2), which mediates transcriptional repression via binding to methylated DNA with localization to heterochromatin [32]. Pathogenic variants in *MECP2* are associated with Rett syndrome spectrum disorders, which range from classic Rett syndrome to mild learning disabilities [33]. Classic Rett syndrome is a progressive neurodevelopmental disorder that primarily affects females and is characterized by apparently normal development in the first six to eighteen months followed by an arrest in development and subsequent regression in language and motor skills [33,34]. Though most *MECP2* disorders result from a de novo pathogenic variant, there are rare cases in which a *MECP2* variant is inherited in an X-linked manner from a heterozygous mother who was asymptomatic or minimally symptomatic due to skewing of X-chromosome inactivation [33]. As a result of this possibility, X-inactivation studies were sent on blood from our patient, and showed no skewing. This variant had previously been reported in two individuals: a 3-year-old female with Rett syndrome who was found to have this variant as part of a complex rearrangement including the *MECP2* gene [35], and in an individual reported to have Rett syndrome without clinical data provided (though the authors reported less severe phenotypes associated with deletions in the c-terminal region) [36]. These variants were not observed at significant frequency in large population cohorts at the time of testing.

We reviewed these results with our colleagues (Dr. Eric Marsh and genetic counselor Holly Dubbs at Children’s Hospital of Philadelphia; Dr. Constance Smith-Hicks and genetic counselors Anna Chassevent and Julie Cohen at Kennedy Krieger Institute). They follow several adults with similar symptoms (significant anxiety, abnormal movements, and progressive distal spasticity). Taken together, while our patient does not have classic Rett syndrome, it does appear that the pathogenic variant in *MECP2* is the cause of her symptoms.

#### Discussion

Genetic Diagnosis & Technology Considerations

Genetic testing is often ordered with a phenotype-first approach, selecting a next-generation sequencing panel for genes known to be related to the condition of interest. This approach, however, does not allow for the discovery of novel genetic causes of that phenotype [37]. Although the analysis pipeline of ES and GS also takes a phenotype-first approach, it permits the discovery of variants of interest, in this case a variant in *MECP2*.

Pathogenic variants in *MECP2* were first identified as causal for young women with Rett syndrome in 1999 [34]. Since then, varied clinical presentations have been reported, including males with cognitive impairment, parkinsonism, pyramidal signs, speech difficulties, brisk reflexes, spasticity, involuntary movements, and psychiatric symptoms [34,38,39], and females with mild intellectual disability, speech difficulties, brisk osteotendinous reflexes, and lower limb spasticity [34]. Given the diversity in phenotypes caused by *MECP2* pathogenic variants, it is likely that *MECP2*-related conditions are under-diagnosed [34]. Our patient’s case expands the field’s phenotypic understanding of this gene: adult women with *MECP2* pathogenic variants may present with a phenotype suggestive of HSP or PLS.

2.Therapeutic Intervention & Personal Impact

The genetic diagnosis enabled the proband to obtain a prescription for a new therapeutic, trofinetide, that was approved for Rett syndrome [40]. She considered enrolling in a clinical trial of copaxone, a multiple sclerosis drug that was evaluated for use in patients with *MECP2*-related conditions in both phase I and II trials (NCT02023424; NCT02153723) [41]. Although the proband did not meet criteria for clinical trials for Rett syndrome, she was able to enroll in a Rett natural history study.

The proband underwent genetic testing with two goals in mind: (1) to clarify the diagnosis, and (2) to provide information to her family about their risk. Genetic diagnosis via ES accomplished both goals.

### 2.5. Case 5

The proband presented for neurologic care at age 53 for evaluation of ataxia. He had developed narcolepsy in his 30s that had only recently been formally diagnosed. He had also developed progressive hearing loss starting in his 40s, followed by optic atrophy. He was experiencing face freezing with strong emotion, attributed to cataplexy. At time of presentation to neurology, the proband reported two to three years of slowed information processing. On neurologic exam, he had decreased sensation to pinprick and temperature in the bilateral distal lower extremities, and EMG demonstrated mild axonal neuropathy. A brain MRI revealed mild atrophy worse in posterior hemispheres, and possible vermian atrophy. He reported that three siblings and two paternal aunts had similar symptoms, and that his father died at age 46 due to cardiac disease and may have had narcolepsy.

Although sending ES plus mitochondrial sequencing as a trio with an affected sibling and unaffected mother was recommended, the testing was ultimately completed with only the proband’s sample. A heterozygous pathogenic variant in *DNMT1* was identified in the patient: c.1709C>T, p.Ala570Val, exon 21, ENST00000359526).

Pathogenic variants in *DNMT1* [OMIM: 126375] are associated with an autosomal dominant neurodegenerative condition comprised of cerebellar ataxia, deafness, narcolepsy, and hereditary sensory neuropathy type 1E, explaining the proband’s phenotype [42]. This gene encodes a DNA methyltransferase maintenance enzyme that contributes to gene silencing [43]. The variant detected in the proband had previously been identified in multiple affected patients, both in de novo cases and in affected families in which the variant has been shown to segregate with disease [44,45], with functional evidence demonstrating that patients with this variant have altered DNA methylation patterns [46].

Secondary findings were also requested as part of this analysis, and two heterozygous variants (one likely pathogenic and one VUS) were reported in *TTN* [OMIM: 188840]. The gene *TTN* encodes titin, or connectin, which is a protein critical to several functions of the cardiac and skeletal muscles [47]. It is unclear whether these variants are in *cis* or *trans*, but cardiac screening was recommended given the association with dilated cardiomyopathy [48].

#### Discussion 

Genetic Diagnosis & Technology Considerations

As described in case examples 2 and 4, ES has the advantage of breadth of coverage. In this case, it was particularly important when evaluating a patient with a complex or evolving phenotype, impacting multiple organ systems. Although gene panels curated for hearing loss and neuropathy include this gene, an ataxia repeat expansion test, which is often the first-tier test sent before or concurrent with exome for ataxia evaluation, would not have detected this finding. Panel-based testing would also not have included the option for secondary findings analysis, which led to the reporting of the *TTN* variants.

2.Therapeutic Intervention & Personal Impact

A genetic diagnosis provided a unifying answer for the proband’s many medical issues, as well as his family’s underlying condition. The proband and his spouse expressed relief at having an explanation for his symptoms, and a better understanding of how his condition related to those of his family members with slightly different presentations. They felt that understanding the cause of his symptoms made them more manageable, especially for the behavior and cognitive changes. A *DNTM1*-specific support group was located through social media, which the proband intended to connect with. This *DNMT1* diagnosis also refined his surveillance/management plan, by adding a sleep specialist and cognitive neurologist to his care and making his cardiologist aware of the *TTN* variant. Additionally, the risk to family members for the neurologic syndrome and potential cardiac problems was clarified, and one of his siblings subsequently underwent genetic counseling and testing.

## 3. Conclusions

Genetic diagnosis is now an integral component of care for adults with NMDs. The examples from our clinics demonstrate how ES or GS impacted patient diagnosis and management. In each example, ES or GS was superior to multi-gene panel testing, including the ability to detect variants in some genes with pseudogenes, to detect repeat expansions, to incorporate parental or other family member samples, and to engage in reanalysis over time. Broad-based testing also helped to identify genetic etiologies in patients with evolving or complex phenotypes, as opposed to reliance on the up-to-date curation of phenotype-based multi-gene panels, which poses a challenge for commercial laboratories given the pace of evolving genetics discoveries. Use of ES or GS additionally enables patients to learn of secondary findings, which may have additional impacts on medical management and family risk. Although an example was not provided here, ES or GS is the main way that novel candidate genes are currently identified [49]. We have also demonstrated that achieving a genetic diagnosis can impact a person beyond ending the diagnostic odyssey: they can receive gene-specific treatments, enroll in research studies, be monitored for disease-associated complications, receive support and other community resources, and provide information to family members. While these benefits are not unique to testing via ES or GS, in each of the cases presented, ES or GS was necessary to arrive at the genetic diagnosis.

Genetic testing technologies are rapidly evolving, and the field is shifting towards greater use of multi-gene testing strategies, including panel testing and ES/GS. Although higher in diagnostic yield than panel testing, ES and GS have limitations as well [50]. For example, some genetic NMDs cannot be detected by ES, such as repeat expansion disorders (e.g., myotonic dystrophy and spinal bulbar muscular atrophy), and neither ES nor GS will capture some disorders with complex genetic variants (e.g., facioscapulohumeral muscular dystrophy) [3]. Future technologies are likely to emerge that address these limitations. Additionally, an inherent limitation to this technology is the frequent lack of data available for variant interpretation. Even when pathogenic or likely pathogenic variants are identified by genetic testing, it may not be immediately clear whether these variants are actually causing the phenotype. Additional analyses can sometimes clarify this: if a variant is being linked to a novel phenotype, one of the strongest validation methods is through functional data in cell or animal models, although this is often not feasible due to the specialized resources, time, and personnel required. Sometimes metabolic screening (such as in patients with MTP deficiency) or enzymatic analysis can yield functional clues to whether the variant is disrupting the gene’s function. There are also accessibility limitations of ES and GS from a cost perspective. These tests can cost several thousand dollars [51], and in the United States, this may be the patient’s direct responsibility if their insurance does not cover the testing. As the cost of sequencing continues to fall, this too is likely to shift in the future [52]. While GS may be more expensive, this test should be prioritized over ES when conditions on the differential are associated with variants not detectable on ES, such as repeat expansion disorders; recommendations for test selection have been described elsewhere [53]. In addition to the barrier of cost, there also exists a barrier in the form of personnel: not all clinics for adults with NMDs will have a genetic counselor or clinician with genetics expertise on staff. Many of the benefits that can be harnessed from ES and GS depend upon having clinicians that can correctly order this testing, interpret the results, and initiate reanalysis over time when appropriate. While ES or GS can be ordered by providers throughout the United States and in many locations around the world, clinical genetics personnel resources vary between and within countries, which influences a clinic’s ability to offer ES or GS. If a provider is comfortable ordering the testing but does not have the expertise to interpret the results, they should be ready to refer to a genetic counselor or tertiary care center with genetics expertise to assist in interpretation of variants.

In addition to the limitations of ES/GS technology and access, there are limitations to the current study worth mentioning. The scope of advantages of broad-based testing we were able to discuss was limited to examples from the families who were able to be reached and chose to provide consent. It is also worth noting that this study was retrospective in design, and cases were purposely sampled to demonstrate key points. Further, although each case selected is itself narrow in scope, our aim was to use these examples to demonstrate elements of broad-based testing that we feel apply across indications.

It is important that providers caring for persons with NMDs not only consider testing beyond multi-gene panels, but also select the broad-based testing that includes coverage of all suspected disorders. ES or GS should be considered either as a first-tier test, or as a secondary test if prior testing is nondiagnostic [3], and can be supplemented by single gene analysis for conditions not captured with exome or genome technology. This case series illustrates that with appropriate use of ES and GS, we can optimize the chances of diagnostic success, and all of the implications therein, for this population of patients.

## Data Availability

Data are contained within the article.

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
