# Peer review of "The Importance of Offering Exome or Genome Sequencing in Adult Neuromuscular Clinics"

_biology, 2024, doi:10.3390/biology13020093_

Round 1
Reviewer 1 Report
Comments and Suggestions for Authors
The well-written study that will be of interest to clinicians and laboratory workers in the area of neuromuscular diseases.
I have a small comment - I would add a gene identification to the individual variants (NM_ possibly other, according to the HGVS nomenclature).
Author Response
We appreciate the reviewer’s kind words. We agree with the recommendation to add transcript information and have added transcript numbers as reported by the commercial laboratory for each sequencing variant presented (this is not applicable for repeat expansion variants).
Reviewer 2 Report
Comments and Suggestions for Authors
This manuscript describes five interesting case reports where the authors describe the added value of using exome/genome sequencing to reach a conclusive molecular diagnosis and to gather a more precise understanding of the patients’ clinical manifestations.
The manuscript is well written and informative. The authors did not discuss the value of extra tests such as functional validation of variants or metabolic screening in the case of patients with altered mitochondrial trifunctional protein. The authors should also comment on the value of constant update of multigene panels (even bringing as example the Panel App initiative of Genome UK). A short comment on the limitations of their study seems also appropriate.
Author Response
We appreciate the reviewer’s compliments. We feel the recommendation to discuss functional validation is excellent and have added this to the discussion. We also have added a comment to the discussion about panel curation, as we agree this is important: gene panels require constant curation, and thus may not be up-to-date at all times with the pace of evolving genetics data. Finally, we now acknowledge limitations of this case series in the discussion.
Reviewer 3 Report
Comments and Suggestions for Authors
Dratch et al., present five rare neuromuscular disease cases where the novel molecular techniques of exome (ES) and genome (GS) DNA sequencing have played critical roles in the diagnosis and, where available, treatment of these disorders. The authors compellingly argue for the much wider use of ES and GS in neuromuscular clinics given their advantages over multi-gene panels, the most used genetic testing approach currently. The specific superiority of ES and GS over gene panels illustrated by these cases included: variant detection in genes with pseudogenes and repeat expansions, greater ability to include samples from family members and to engage in reanalysis. Limitations of ES and GS were also discussed in the Conclusions, while the fact that multi-gene panels are more affordable and available was mentioned briefly in the Introduction.
It would be informative to clarify if ES/GS sequencing and downstream bioinformatic analysis was done entirely or in part by commercial vendors.
Most of the authors are affiliated with a top-tier medical center, where some of the necessary technical resources are likely available on site. This may not be the case for the majority of public or private clinics that may deal with these kind of patients. More specific suggestions in the Conclusions on how to proceed for the latter providers, so that they make available these technologies for diagnosis of their patients would be helpful.
Author Response
We thank the reviewer for their close read of our manuscript. We appreciate these comments which we feel strengthen/add clarity to the manuscript. All genetic testing (ES/GS sequencing, interpretation, and reporting) was completed at commercial CLIA-certified laboratories as is the standard for clinical genetics care in the USA; we now state this at the end of the introduction where other methods-style information is presented. We also agree that resources may vary between and within countries, and now elaborate on the issue of resources with advice for providers in the discussion.
Reviewer 4 Report
Comments and Suggestions for Authors
Author Response
1. We agree with the reviewer that in this journal it would be appropriate to include more description of the function and biology underlying each altered gene, and have updated the manuscript along these lines in response to this reviewer’s first and third points, which has strengthened the manuscript significantly. With regards to the terms WES and WGS, as we wrote in our response to the editor, these terms are technically inaccurate as neither the sequencing nor the analysis are technically “whole,” which is why we elected to use ES and GS. We have added a sentence about this in the introduction along with references.
2. We appreciate these thoughtful comments.
3. We have added the OMIM number for the gene associated with the diagnosis in each case example. We have also added information about the function of the gene products. We appreciate the desire to have more information included about the frequency of the variants and have added some frequency information for variants when possible, such as whether variants had previously been reported in the literature. All testing was completed at commercial CLIA laboratories (which we now state explicitly in the manuscript), which share their variant data with ClinVar; therefore we feel that including the fact that this is present in ClinVar does not add additional value to the manuscript. We also feel that including more specific information from gnomAD on frequency is not relevant (these are controls), and instead comment on the lack of significant frequency when relevant.
4. All of the commercial laboratories utilized for testing are CLIA-certified laboratories, which we now indicate at the end of the introduction. We have chosen to not report the names of the commercial laboratories, as this could be seen as an endorsement of those laboratories. Commercial laboratories use different pipelines for analysis and variant filtering, but all use Richards et al. (2015) ACMG classification for interpretation (https://doi.org/10.1038/gim.2015.30 ), which we now state explicitly.
5. We thank the reviewer for suggestions the addition of these SORD and ATXN2 references, we now include them in the text where appropriate. Although the DNMT1 reference says “books” in the URL, it is from the GeneReviews resource which is available online to readers in the same way that other scholarly articles are, and therefore we elected not to remove this reference.
6. We appreciate the reviewer’s attention to this topic of health economics, which often makes practical application of ES and GS challenging. We do acknowledge in the introduction that testing strategies other than ES or GS have historically been utilized due to cost constraints. We have added more information to the discussion about some of the economic and resource constraints and now clarify that cost of testing, not just insurance coverage, poses a barrier. We omit details on exact prices of ES and GS for patients, as even within the US this is highly variable between labs and between patients undergoing the same test with different insurance plans. These prices are also rapidly changing which we acknowledge in the discussion.
7. We appreciate the reviewer’s comment about the phrase “learning differences,” but we elect to keep the word “differences” as opposed to “difficulties,” as this is how the patient described her learning experiences. She was not unable to complete tasks, but rather processed information differently than her peers, and we would like to stay faithful to how she described her experiences.
Round 2
Reviewer 2 Report
Comments and Suggestions for Authors
The authors replied correctly to my comments
Comments on the Quality of English Languagenone